# How does reorganisation in child and adolescent mental health services affect access to services? An observational study of two services in England

**Mina Fazel**[1]*, **Stephen Rocks**[2], **Margaret Glogowska**[3], **Melissa Stepney**[3], **Apostolos Tsiachristas**[2]

1 Department of Psychiatry, University of Oxford, Warneford Hospital, Oxford, United Kingdom, 2 Nuffield Department of Population Health, Health Economics Research Centre, University of Oxford, Oxford, United Kingdom, 3 Nuffield Department of Primary Care Health Sciences, University of Oxford, Oxford, United Kingdom

* mina.fazel@psych.ox.ac.uk

**Data Availability Statement:** The CAMHS quantitative data are available from Oxford Health NHS Foundation Trust for researchers who meet

## Abstract

### Background

Child and Adolescent Mental Health Services (CAMHS) in England are making significant changes to improve access and effectiveness. This 'transformation' variously involves easier access to services through a Single Point of Access (SPA), more integrated services within CAMHS and enhanced co-provision across education and third sector or non-profit organisations.

### Methods

A mixed-methods observational study was conducted to explore the process and impact of transformation over four years in two services. Ethnographic observations and in-depth interviews were conducted and Electronic Patient Records with over one million contacts analysed. Difference-in-differences analysis with propensity score matching to estimate the causal impact of the transformation on patient access was utilised.

### Outcomes

Spend and staffing increased across both CAMHS. The SPA had growing rates of self-referral and new care pathways were seeing patients according to expected degree of psychopathology. Third sector partners were providing increasing numbers of low-intensity interventions. Although the majority of staff were supportive of the changes, the process of transformation led to service tensions.

In the first year after transformation there was no change in the rate of new patients accessing services or new spells (episodes of care) in the services. However, by year three, the number of new patients accessing CAMHS was 19% higher (Incidence Rate Ratio:

the criteria for access to confidential data. Oxford Health data can be used according to the following statement/condition from their DSA: "Data access is restricted to researchers using the data for pre-defined purposes". It would therefore be illegal to make the data public under the DSA that is the condition of data access. Interested parties can apply to get access to the data from Oxford Health NHS Foundation Trust on direct application to the following email: cris.admin@oxfordhealth.nhs.uk and by following the procedures as outlined in the following website: https://www.oxfordhealth.nhs.uk/research/toolkit/cris/. The study data was accessed as part of a data sharing agreement with the University of Oxford. These restrictions exist because these are identifiable electronic patient records and therefore contain potentially identifying or sensitive patient information. The anonymised qualitative interview transcripts are available by writing to databank@psych.ox.ac.uk. This data falls within ICO guidelines health data as falling under special category data that needs additional care. 'Special Category data' therefore cannot be made publicly identifiable as this was not in the explicit consent process, however, specific requests will be accommodated and data provided according to research questions. We conducted a small number of interviews, the services are well aware of the individuals who were interviewed and their responses can therefore be easily identified by those within the service, compromising their confidentiality.

**Funding:** This research was funded by the National Institute for Health Research (NIHR) Applied Research Collaboration Oxford and Thames Valley (MF, AT, SR, MS, MG). https://www.arc-oxtv.nihr.ac.uk The views expressed in this publication are those of the author(s) and not necessarily those of the NHS, the NIHR or the Department of Health and Social Care. Additional funding came from the Clinical Commissioning Groups in Oxfordshire and Buckinghamshire. The funders had no role in study design, data collection and analysis, decision to publish, or preparation of the manuscript.

**Competing interests:** The authors have declared that no competing interests exist.

1·19, CI: 1·16, 1·21) and the rate of new spells was 12% higher (Incidence Rate Ratio: 1·12, CI: 1·05, 1·20).

## Interpretation

Transformation investment, both financial and intellectual, can help to increase access to CAMHS in England, but time is needed to realise the benefits of reorganisation.

## Introduction

There is heightened interest across many countries as to how mental health services can best meet the needs of their child and adolescent populations [1,2]. The rising number of young people presenting with mental health needs is propelling services, across a number of nations, to introduce broad systemic changes. Making child and adolescent mental health services (CAMHS) more accessible to young people is driving many changes taking place in England, which have been mirrored in countries including Canada, Australia, the US and Ireland [3–8]. Access to services for children and adolescents include a number of different factors, including: knowledge, attitudes, and beliefs about mental health problems and treatment held by both young people and their parents; where these services are geographically located; how they can be approached; and once approached whether they have the resources to address any identified need [9].

In England the focus on improving access to services, for those under the age of 18, has led to a major reorganization of CAMHS aligned with the THRIVE model which recommends principles by which services can be organized [10] emphasising five areas: Thriving, Getting advice and signposting, Getting Help, Getting More Help and Getting Risk Support. The focus of this model lies in making the experience of mental health services for the young person less confusing and encourages shared decision-making with the children, adolescents and their families. The new model tries to enable all the agencies that are involved in a young person's life to work together in a coherent and integrated manner.

Government directives have therefore encouraged many CAMHS services, through commissioners, to 'transform' their services to improve accessibility, increase quality of care and improve health outcomes [4,11], although the pace and direction of change is determined at the local level. Therefore a natural experiment is taking place across CAMHS in England where different components of services are being changed to try and better address the identified needs of the young population and improve access to services in general. Many of the services have even adopted the THRIVE terms for the different parts of their newly 'transformed' services.

This observational mixed methods study aims to build on the limited evidence-base that currently exists on the impact and implementation of these system wide changes to CAMHS. We will describe components of two CAMHS services experiencing service 'transformation' at different time points and with some differences in how the changes are translated and evaluate their resultant changes in access to services, as measured by the actual number of children and adolescents entering the services.

## Methods

### Study setting

The study was conducted in CAMHS provided by Oxford Health NHS Foundation Trust (Oxford Health) (Supporting information 1: Map and Demographics of study area in S1 Appendix). Oxford Health is one of the largest CAMHS providers in England [12], delivering services for Clinical Commissioning Groups (CCGs) to NHS services in Buckinghamshire (Bucks); Oxfordshire (Oxon); and Swindon, Wiltshire, and Bath and North East Somerset (SWB). We investigated the CAMHS changes or 'transformation' in Bucks and Oxon. These services are commissioned locally and therefore we were able to examine the changes taking place in the different areas as described in the study protocol [13]. The changes in Bucks took place from 2015 and in Oxon from 2018 as each area made changes according to when their service agreements are being renegotiated and designed. For many services these take place on a five-year cycle and therefore depending on when these are confirmed, the new services were introduced.

### Design and research methods

We followed an implementation science approach utilising mixed-methods as detailed in the study protocol [13] with the Reach, Effectiveness, Adoption, Implementation and Maintenance (RE-AIM) framework. This framework has a particular focus on implementation of complex interventions in real-world settings [14].

This study aimed to understand the implementation of CAMHS transformation and assess its impact on access to care by addressing the following research questions:

1. What components are being adopted as part of the CAMHS transformation? (Adoption)

2. How are the transformations being implemented and maintained? (Maintenance)

3. What are the facilitators and barriers to transformation? (Implementation)

4. What is the effect of transformation on service activity / pathways? (Reach)

5. What is the impact of the transformation on patient access? (Effectiveness)

**Qualitative data and analysis.** Qualitative data collection (by MS, January 2018 to March 2019) focused on the CAMHS transformation in both Bucks and Oxon including 80 hours of ethnographic observations—shadowing key staff, informal interviews with different stakeholders and attending team meetings. Observations were determined by access to particular settings (such as the SPA and meetings) through key staff or 'gatekeepers'. A field diary was kept throughout to record observations with particular attention paid to the RE-AIM framework. The objective was to record and observe naturally occurring interactions: watching, listening, and better understanding how transformation processes were occurring and being interpreted 'on the ground'. The observation period for this study included both a pre- and post-transformation period. In addition, in-depth interviews with eighteen members of staff, including a range of administrative and clinical staff, service managers, team leads and those involved in the Single Point of Access (SPA), specialist pathway services and third sector (charity, social enterprises and voluntary group) partners. The questions asked were framed around the implementation questions of interest, including what they perceived as the main changes taking place in their organisations, what they thought the benefits and problems were associated with these changes and the processes by which these changes had been introduced and sustained. Interviews were also conducted with eight young people and four parents/carers (2

paired interviews) with experience of using CAMHS pre- and post-transformation. All the interviews were digitally recorded and professionally transcribed. Thematic analysis of the data followed a modified grounded theory approach using N-Vivo [15].

**Quantitative data and analysis.**   Data from electronic patient records was available from April 2012 to March 2019, encompassing pre- and post-transformation. We also had access to all routine outcome measures (ROMs) recorded electronically over this period, including the Revised Child Depression and Anxiety Scale (RCADS) which is a well-validated and widely used measure that gives a score denoting clinical caseness. Data on whole-time equivalents and direct staff spend was available from 2013/14 to 2018/19, with whole CAMHS funding available from 2015/16.

Data was grouped into distinct spells of care (care received from admission to discharge) and analysed from October 2013 to March 2019 [16]. A quasi-experimental approach to evaluating this 'complex' intervention was adopted [17,18]. A difference-in-differences (DiD) framework was employed and supplemented with propensity score matching for the analysis of re-referrals and waiting times [16]. In the main analysis, Bucks was compared with Oxon. For a sensitivity analysis, both Oxon and SWB were included as the control group.

We adopted a parallel convergent mixed methods design [19]. The quantitative and qualitative data were collected concurrently and analysed separately. We then integrated the quantitative and qualitative datasets to address the research questions in the following way. Hypotheses (arising from the quantitative data) and themes (from the qualitative data), generated from the separate analyses, were explored in the light of the findings from the other dataset. In this way, key issues emerging from one of the components was then examined alongside evidence from the other component. This enabled us to obtain explanation and context around an issue, as well as prompting new questions to take back to the other dataset. This approach, known as 'following a thread' (described by O'Cathain et al) [20], which involves iteratively moving between quantitative and qualitative datasets, enabled us to achieve multiple perspectives on the transformation and more complete interpretation and understanding of the impacts of the reorganization.

**Ethics.**   Anonymized routinely collected patient level data for this study was shared according to a local data sharing agreement with Oxford Health NHS Foundation Trust. This study was registered with Oxford Health NHS Foundation Trust approved by the Children and Young People's directorate (4/4/2018). Interview participants gave informed consent for participation.

## Results (including refined methods)

### Components of CAMHS transformation

Bucks and Oxon had similar core components of transformation which are each reported below, namely:

- a single point of access (SPA) to CAMHS with the possibility for self-referral;

- new service delivery pathways and other specialist teams;

- third sector partners directly commissioned to deliver services with or as part of CAMHS.

The transformations in Bucks and Oxon were aligned with the transformations taking places across England, as evidenced by reviewing a local transformation plan from each NHS region (the first alphabetically placed commissioning group) (Fig 1). The majority of plans included a SPA with the possibility of self-referral, many used the THRIVE model and all included third sector involvement.

| Clinical Commissioning Group | NHS Region | Components of transformation | | | |
|---|---|---|---|---|---|
| | | SPA, incl. self-referral into CAMHS | Thrive model | Commission staff to act as link with schools | Direct commissioning third sector |
| Barking and Dagenham | London | ✔ | | ✔ | |
| Berkshire West | South East | ✔ | ✔ | ✔ | ✔ |
| Birmingham and Solihull | Midlands | ✔ | | | ✔ |
| Bolton | Greater Manchester | | ✔ | ✔ | |
| Bradford | Midlands | | | ✔ | |
| Bristol | South West | | ✔ | | ✔ |
| Cheshire | Cheshire and Merseyside | | ✔ | ✔ | ✔ |
| Darlington | NE and Yorkshire | ✔ | ✔ | | |
| Devon | South East | | ✔ | | |
| Kent | South East | ✔ | ✔ | | |
| Lancashire | Lancashire and South Cumbria | | ✔ | ✔ | |
| Northants | Midlands | ✔ | ✔ | ✔ | ✔ |
| South Essex | East of England | ✔ | | n/a | |
| Staffordshire | North Midlands | ✔ | ✔ | | |

**Fig 1. Components of transformation across CAMHS in England.**

**Funding for transformation.** The funding allocated by each commissioning group to the studied CAMHS from 2015 to 2019 is included in Supporting information 2 in S2 Appendix. As described elsewhere, we excluded inpatient spend and adjusted for the population [21]. Specifically, we estimated the target population by multiplying the population aged under 18 by the estimated prevalence of common mental disorders in that age group. Spend per head of target population increased 7% in Bucks between 2015/16 and 2018/19; in Oxon spend increased 22% over the same period.

The wider context for transformation is rising demand for CAMHS. NHS England set a performance target of 35% of the population in need being seen by CAMHS (two contacts in one year, excluding SMS and email) [11]. Performance against this target improved in both Bucks and Oxon with a steady increase to 35% in Bucks (2018/19) and in Oxon, the rate peaked at 44% in 2017/18, before falling to 41% in 2018/19 (Fig 2).

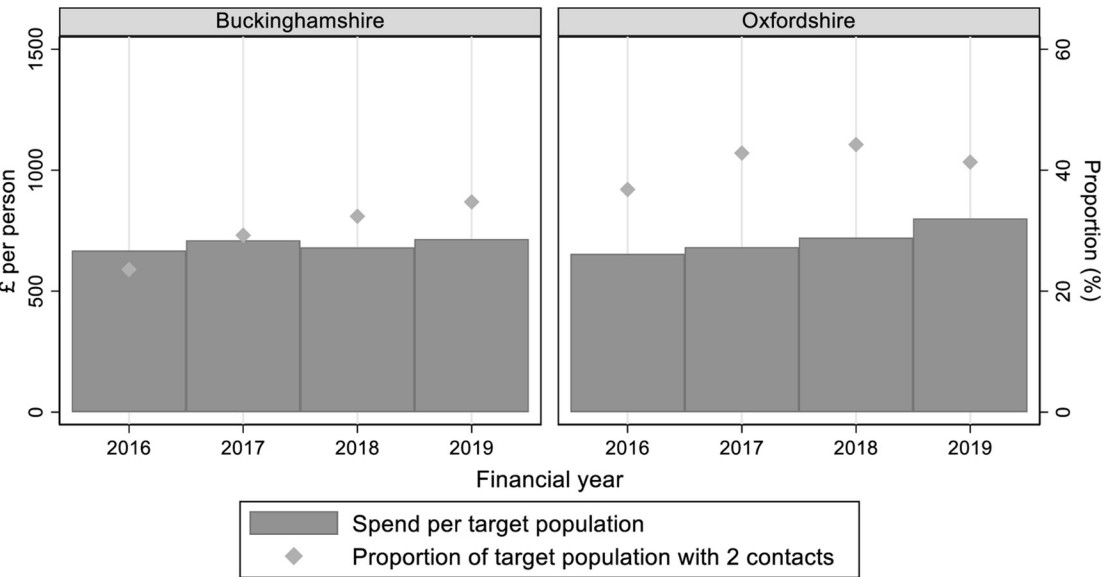

**Fig 2. CAMHS budget and performance against attendance target.**

**SPA.** For staff, the introduction of the SPA was perceived as the core change of the CAMHS transformation, with a central team based in one location taking online and telephone referrals from health, social care and education professionals, family members, as well as young people themselves. This was evident from the redesign of how the services were being managed, the staff interviews and the observations from the services. A particular innovation was the ability of parents/carers and young people to self-refer (primarily for those aged 16 and 17, but available to younger callers). This component was designed to improve accessibility and rationalise referrals.

"…anyone can access us and have a conversation as soon as they have got a question about mental health and we will listen to them and try and do something with that…For me, the biggest single change that we have managed to bring about lies in SPA."

(*Staff interview*)

In both Bucks and Oxon, the numbers of people accessing SPA climbed quickly following initial awareness-raising activities [22]. While much of the emphasis on SPA was on the phone line, as the service evolved, increasing numbers of requests were received online, as described in more detail by Rocks et al [22] The introduction of the SPA was associated with an increase in self-referrals, both from carers and children and adolescents, and a fall in the proportion of referrals from primary care [22].

For some young people and families we interviewed, it was important that the SPA change was clear and that the new service provision had clarity:

"(*Young person*) So it's a referral line, really? Would it do anything [..]so like if a teenager phoned up who is using it kind of like Samaritans type call, does it have that—[interviewer explains]

*(Parent) It needs to be clear what it is, doesn't do or what it isn't at least.*"

**New pathways.**   The CAMHS transformation involved a change from a services that were separated into four different tiers to a more integrated approach to the different services. For example, the tiered approach to services had children and adolescents seen in the service tier that corresponded with their needs- with tier one services often delivering services in the community and tier three and four services for children and adolescents with more severe or complex mental health difficulties likely needing a multidisciplinary and more specialized approach. These were changed in the THRIVE model to 'pathways' [10,23] called 'Getting Help' and 'Getting More Help', where the young person would try and be seen for all their needs within that team and then with additional specialty teams in the services, for example for children and adolescents with eating disorders and neurodevelopmental conditions.

Fig 3 compares the breakdown of activity across the new pathways from Electronic Patient Record (EPR) data alongside the staffing mix in terms of whole time-equivalents. Prior to transformation, the majority of CAMHS contacts were in tiers 2 (Bucks 32%; Oxon 23%) and tiers 3 (Bucks 45%, Oxon 42%) (with tier 3 providing services for cases with greater complexity). Following the introduction of the new CAMHS pathways, 21% (Bucks) and 23% (Oxon) of patients were seen in 'Getting Help' and 26% (Bucks) and 24% (Oxon) in 'Getting More Help'.

"*As the service has become more accessible the amount of referrals and the amount of work coming in for [Getting Help] has just mushroomed.*"

(*Staff interview*)

The number of contacts per spell for each pathway suggests Getting Help and Getting More Help are used appropriately, with more intensive support from the latter (S3(i) Appendix). The initial severity of patients measured by the RCADS was higher among those first entering Getting More Help than Getting Help (S3(ii) Appendix).

**Third sector partnerships and improved working with education.**   In Bucks, third sector involvement was primarily with Barnardo's, one of the largest charities providing care to children in England. The Barnardo's workers were placed into SPA as Contact Support Workers (CSW) where they would take initial calls and liaise with clinicians about referral decisions; and into the Getting Help pathway as 'Buddies' providing low-intensity interventions (S3(i) Appendix). The Getting Help pathway had seen almost 4,400 unique patients–the most of any pathway—and delivered more than 34,000 contacts (appointments) between 2015/16 to 2018/19. Staff perceived that demand for this pathway had grown. However, there was tension between being able to meet the perceived needs of an individual and the pressure to see others waiting for care as the numbers trying to access the service increased.

In Oxon, 'Community InReach' was established to enable CAMHS to work more closely with eight third sector partner organisations to provide alternative and supplementary support for young people.

"*. . .a lot of those charities naturally engage with a group that we have always struggled to engage with.*"

(*Staff interview*)

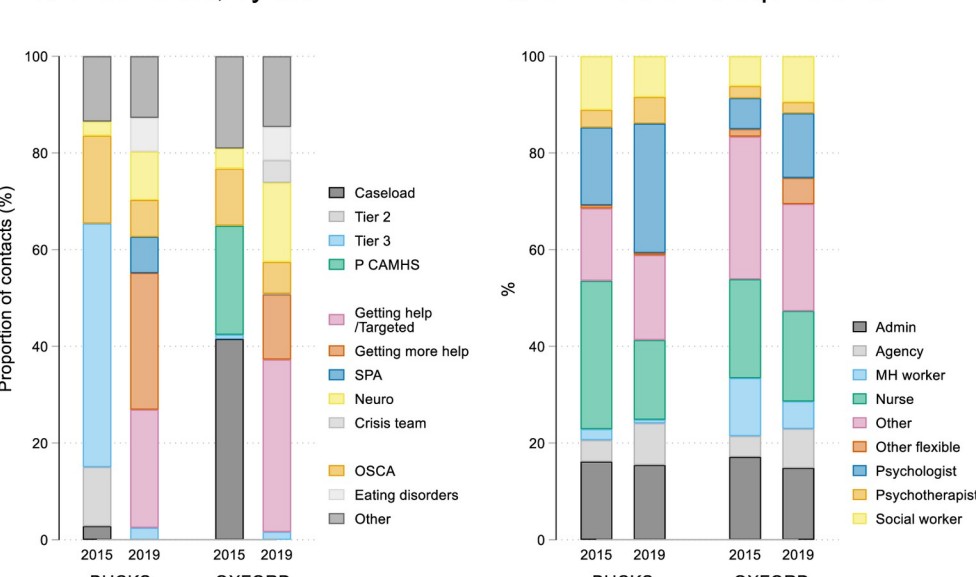

**Fig 3. Proportion of patients seen by team and their whole-time equivalents.**

These third sector partnerships had only become fully operational at the end of the study period.

In Oxon, mental health staff provide an 'InReach' service to secondary schools offering advice, support and training with an emphasis was on giving school staff and students specific tools to respond to mental health problems, such as symptoms of anxiety, through training, workshops, assemblies and group work.

## What are the facilitators and barriers to transformation?

**Facilitators.** *Shared vision for transformation.* The majority of staff in both services who were interviewed were positive about the vision for CAMHS transformation.

"*My instinct was, it was the right model with the right language with good principles and good thinking behind it and some evidence.*"

(*Staff interview*)

In part, this reflects perceived difficulties with the previous model of CAMHS and a recognition that change was needed:

". . .*it was on its knees. I think, anyway.*

(*Staff Interview*)"

Staff in both services also felt the overall direction of the respective transformations were contributing to the development of a better service. Staff saw SPA as a way of ensuring that those children and adolescents coming into CAMHS really needed specialist help, but that others who did not, still received support and guidance.

*Staff commitment*. Staff showed determination to deliver services aligned with the new vision for transformation, and although there were many changes that they had to manage, many described support for the overall programme of work:

"*My instinct was, it was the right model with the right language with good principles and good thinking behind it and some evidence*"

(*Staff Interview*).

They agreed with the overall vision:

"*I think it's creative. I think it's much more kind of proactive. . .there is a lot more emphasis on trying to work in the community and the proactive kind of looking at the early signs to really try and support our colleagues. . .in more preventative type kind of measures.*"

(*Staff Interview*)

Furthermore, staff demonstrated their commitment to the SPA with statements such as:

"*it's just much more of a friendly front door to the service.*

(*Staff Interview*)

". . .*you can ring in at any time and you can just be reopened. You don't need a letter. You don't need a form. You don't need anybody else to do it for you. I think that's a big plus.*"

(*Staff Interview*) [22]

"*We are meant to do two days a week in SPA, but because we've had so many calls. . .If somebody else is on it and I know they're inundated, I'll say 'I'll do some of them'. . .And definitely probably work an extra hour at least each day, and don't take a lunch.*"

*(Staff interview)* [22]

**Barriers.**   *Destabilising a strained system*. For many of the staff in both services, the CAMHS transformation was perceived as initially destabilising, with staff shifting into new posts or waiting to hear about their new roles or contract renewal, while other staff left the service. In both services, the transformation took place in less than 6 months with many staff not feeling they fully understood what would be happening. Some staff reported initial difficulties, working both within the new and old system. Across the CAMHS transformations, many of the staff felt that the implementation of the changes had been impeded by overall lack of staff. CAMHS had an increased reliance on agency staff (Fig 3), which was, however, less pronounced within the third sector partners. A key finding from the interviews with young people and parents was the perceived lack of staff consistency in CAMHS.

*One of the reasons why my mum thought I relapsed again was because I had so many different people in my care at CAMHS. There was none of that consistency. Obviously, I understand that people had to move on. There was never a consistent person that wanted, well it felt to me like they wanted to help me.*

(*Interview, young person*)

*Challenges around third sector involvement.* Third sector involvement in CAMHS was new and brought challenges, including agreeing standard operating procedures, creating information governance systems, defining roles, and building familiarity. In Bucks, at the start of the transformation there were comments that there were 'differences in organisational and working culture' between the organisations.

"*the call handlers were [third sector] staff. They had just come into mental health and into the health service. So for them it was working under the NHS. . .So the process, the governance, the previous experiences would be very different. So for them, it was a big leap into this massive organisation*"

(*Staff interview*)

### Effect of transformation on service activity

Bucks, which transformed in 2015/16, was compared with both Oxon and SWB (as controls), which did not transform until 2018/19. Table 1(A) provides the results by year following transformation. Relative to the control services, in Bucks one year after the transformation there was no significant difference in the rate of new spells (Incidence Rate Ratio: 1.17, CI: 0.95, 1.44) or new patients (Incidence Rate Ratio: 1.17, CI: 0.95, 1.44) accessing the service. Performance improved relative to the control services in subsequent years. There were significant increases in both in years 2 and 3. By year 3, the rate of spells was 19% (Incidence Rate Ratio: 1.19, CI: 1.16, 1.21) and new patients 12% higher (Incidence Rate Ratio: 1.12, CI: 1.05, 1.20). The re-referral rate–those referred back into the service at least one month after discharge–in Bucks was not significantly different from the control services in any of the three years (S3(iii) Appendix).

Examining Waiting times, in year 1, there was a relative increase in waiting time for the first contact of 13% (Incidence Rate Ratio: 1.13, CI: 1.05, 1.21) compared to the pre-transformation period. By year 3, waiting times for the first contact were 21% lower (Incidence Rate Ratio: 0.79, CI: 0.73, 0.85). There was no significant difference in waiting time for the second contact in year 1 (Incidence Rate Ratio: 1.02, CI: 0.93, 1.12) or year 3 (Incidence Rate Ratio: 0.96, CI: 0.88, 1.06), but this increased in year 2 (Incidence Rate Ratio: 1.12, CI: 1.03, 1.23) (S3 (iv) Appendix).

**Sensitivity analysis.** In the sensitivity analysis, Bucks, which transformed in 2015/16, was compared with Oxon alone, which did not transform until 2018/19. Table 1(B) provides the results by year following transformation, which support the primary analysis.

### Discussion

Broad system changes are taking place across child and adolescent mental health services in the UK and this study describes a methodology for how to evaluate changing CAMHS services with a focus on access to services. Both services had increased financial investment and with the new referral pathways, increasing numbers of patients were contacting the SPA directly with an increase in 19% of new patients accessing the service by year three. The impact of the changes in Oxon and Bucks on improving access to mental health services were not immediately evident but became more apparent over time, consistent with research into other large-scale transformations in the NHS [24,25]. Waiting times for the first contact were not significantly different in year two, but fell significantly in year three. Patients seemed appropriately triaged into identified care pathways, with more severe cases in the Getting More Help

**Table 1. Impact of CAMHS transformation in Bucks-main & sensitivity analysis.**

**Panel A: Main analysis-Buckinghamshire vs. Oxfordshire and SWB**

| | Rate of new spells per month (per 1,000 under 18) | Rate of new patients per month (per 1,000 under 18) | Likelihood patient already seen (previous spell) | Waiting time for first contact | Waiting time for second contact |
|---|---|---|---|---|---|
| **Year 1** | | | | | |
| DiD, 95% CI, | 1.17 [0.95,1.44] | 1.17 [0.96,1.44] | 1.02 [0.88,1.19] | 1.13*** [1.05,1.21] | 1.02 [0.93,1.12] |
| SE | (0.12) | (0.12) | (0.07) | (0.04) | (0.05) |
| **Year 2** | | | | | |
| DiD, 95% CI, | 1.20*** [1.13,1.27] | 1.12** [1.08,1.16] | 1.02 [0.88,1.18] | 1.06 [0.98,1.14] | 1.12* [1.03,1.23] |
| SE | (0.04) | (0.02) | (0.07) | (0.04) | (0.05) |
| **Year 3** | | | | | |
| DiD, 95% CI, | 1.19*** [1.16,1.21] | 1.12*** [1.05,1.20] | 1.11 [0.97,1.28] | 0.79*** [0.73,0.85] | 0.96 [0.88,1.06] |
| SE | (0.01) | (0.04) | (0.08) | (0.03) | (0.04) |
| Observations | 162 | 162 | 40757 | 38064 | 31541 |
| Family | Poisson | Poisson | Binomial | Poisson | Poisson |
| Link | Log | Log | Logit | Log | Log |

**Panel B: Sensitivity Analysis-Buckinghamshire vs. Oxfordshire**

| | Rate of new spells per month (per 1,000 under 18) | Rate of new patients per month (per 1,000 under 18) | Likelihood patient already seen (previous spell) | Waiting time for first contact | Waiting time for second contact |
|---|---|---|---|---|---|
| **Year 1** | | | | | |
| DiD, 95% CI, | 0.93 [0.81,1.06] | 0.93 [0.83,1.06] | 1.01 [0.86,1.20] | 1.12** [1.04,1.20] | 1.03 [0.92,1.14] |
| SE | (0.06) | (0.06) | (0.09) | (0.04) | (0.06) |
| **Year 2** | | | | | |
| DiD, 95% CI, | 1.08 [1.00,1.17] | 1.01 [0.92,1.11] | 1.02 [0.87,1.20] | 1.01 [0.93,1.09] | 1.12* [1.01,1.24] |
| SE | (0.04) | (0.05) | (0.08) | (0.04) | (0.06) |
| **Year 3** | | | | | |
| DiD, 95% CI, | 1.09*** [1.07,1.12] | 1.08*** [1.06,1.10] | 1.04 [0.89,1.22] | 0.69*** [0.64,0.75] | 0.95 [0.85,1.04] |
| SE | (0.01) | (0.01) | (0.08) | (0.03) | (0.05) |
| Observations | 108 | 108 | 27456 | 25273 | 21652 |
| Family | Poisson | Poisson | Binomial | Poisson | Poisson |
| Link | Log | Log | Logit | Log | Log |

***p-value<0.001,

**p-value<0.05, and

*p-value<0.01

DiD: Difference-in-Differences estimation.

pathway compared to Getting Help where the interventions were also shorter in duration. Third sector partners were also providing increasing numbers of low-intensity interventions.

The majority of staff were positive about the vision of the transformation, and willing to expend considerable effort to see it work, but communication and staffing emerged as barriers to transformation, alongside challenges in recruitment and staff turnover. Staffing shortages potentially provide impetus for partnership working to expand the workforce.

There are only a limited number of previously published studies examining broad changes to community-based child mental health services. Although implementation research has

identified promising strategies to improve services including: enhanced engagement to retain families in services [26]; improved training and support for evidence-based practices [27]; and expanded measurement and feedback systems to monitor services in real time, actual evaluation data remains limited [28]. The only studies identified in the last decade that have examined broad system change to improve demand and capacity in child and adolescent mental health services have included studies on the positive impact of introducing a 'Choice and Partnership' Approach. These services conduct an initial appointment to reach a shared understanding of patient and family needs with a range of options offered following the meeting [29–31]. Another study evaluated 'Shared care mental health care' (with primary care) [32] demonstrating how this model increases access to care as well as decreasing demand on services and a final study of integrated behavioural health care into primary health care systems showed some positive improvements in symptoms [33]. There are therefore some broad similarities with the current 'transformation' of CAMHS in England with a focus on integration and better shared decision-making.

There have been a number of evaluations of youth mental health service change for youth aged up to 25 years, in particular of the Australian headspace initiative [34–37]. Although relevant, the target age range is different. Numerous studies describe specific interventions introduced into child mental health systems, including: family check-up [38]; wraparound services for serious emotional disturbance [39]; parenting interventions [40]; development of interim services whilst awaiting mental health services [41]; free counselling support [42]; telephone-based treatments [43]; assertive outreach teams [44]; and early intervention in psychosis services [45,46].

A few articles have been published on the recent, broader system changes in the UK, primarily on the introduction of the Children and Young People's Improving Access to Psychological Therapies (CYP-IAPT) changes [47]. These are changes that have preceded the current CAMHS transformation, and have focused on ensuring that there are both more CAMHS practitioners and that these practitioners are trained to deliver evidence-based psychological therapies to child and adolescent populations [48]. The studies reporting on these changes do share similar findings on operational difficulties, the need for stakeholder involvement and the importance of leadership, although actual service-related measures, such as in this study, have not been reported [49,50]. The current CAMHS transformation has been described in some publications [51] with qualitative work conducted on key stakeholders [52], although some concerns raised that insufficient young people and parents have been included [53]. Quantitative data on the impact on access to services and efficiency in newly 'transformed' CAMHS service delivery has not been identified in any published studies to date- reflecting the recency of these changes. This study is therefore a timely evaluation of CAMHS provision in the context of transformation.

Strengths of this study include the use of data across one large NHS Foundation Trust, with different services, there was a consistent recording system, increasing the comparability of the data. Untreated mental health conditions negatively impact on development throughout the life course, making access to 'adolescent-responsive' and high-quality health systems crucially important [54] and the need to think about bringing services together in more integrated pathways of care [55] as well as finding ways to prioritise the student voice in how these services are organized [56].

## Limitations

Limitations of the study include the timing of the analysis as the qualitative research was retrospective in Bucks. As this study is a snapshot in time, the changes taking place are ongoing and

subject to flux. We were also unable to draw on sufficient Routine Outcome Measures to comment on the overall effectiveness of the transformation (because insufficient numbers were available to analyse) and we did not interview staff who had left the service. For the estimates of impact, we applied techniques appropriate to the non-experimental design, but could not entirely eliminate the potential for residual confounding [57]. We were also limited in the number of available comparator groups; with a greater number of CAMHS as controls, other techniques such as synthetic matching would permit matching on outcomes in the pre-period [58]. Finally, the intervention took place in a relatively affluent part of England, raising questions as to the extent to which results can be extrapolated to other CAMHS.

## Conclusions

There is pressure to improve child and adolescent mental health services across many nations. Our findings provide robust evidence on how major service changes to child mental health services can be evaluated. Such service evaluations are rarely conducted and published because of the complexity of the intervention considered. This study highlights how the model of service change or 'transformation' adopted across many English services can help to increase access to services, but it takes time for the benefits of reorganisation to be realised.

## Supporting information

**S1 Appendix. Location and demographic characteristics of the study setting.**
(TIF)

**S2 Appendix. Clnical Commissioning Group (CCG) funding allocated to Buckinghamshire and Oxfordshire CAMHS.**
(TIF)

**S3 Appendix. Detailed quantitative results.**
(TIF)

## Author Contributions

**Conceptualization:** Mina Fazel, Apostolos Tsiachristas.

**Data curation:** Stephen Rocks, Melissa Stepney.

**Formal analysis:** Stephen Rocks, Margaret Glogowska, Melissa Stepney.

**Funding acquisition:** Mina Fazel.

**Investigation:** Stephen Rocks, Margaret Glogowska, Melissa Stepney.

**Methodology:** Mina Fazel, Stephen Rocks, Melissa Stepney, Apostolos Tsiachristas.

**Supervision:** Mina Fazel, Margaret Glogowska, Apostolos Tsiachristas.

**Writing – original draft:** Mina Fazel, Stephen Rocks, Margaret Glogowska.

**Writing – review & editing:** Mina Fazel, Stephen Rocks, Margaret Glogowska, Melissa Stepney, Apostolos Tsiachristas.

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
