## [Decision Letter · Decision Letter 0]

1 Feb 2021

PONE-D-20-38869

How does reorganisation in child and adolescent mental health services affect access to services? An observational study of two services in England

PLOS ONE

Dear Dr. Fazel,

Thank you for submitting your manuscript to PLOS ONE. After careful consideration, we feel that it has merit but does not fully meet PLOS ONE’s publication criteria as it currently stands. Therefore, we invite you to submit a revised version of the manuscript that addresses the points raised during the review process.

One of the prior reviewers has completed this review and recommended major decision. I agree with the suggestions made by the reviewer and hope that these will help you in preparing the revision.

We look forward to receiving your revised manuscript.

Kind regards,

Chung-Ying Lin

Academic Editor

PLOS ONE

Journal Requirements:

4. We note that Figure S1 in your submission contain map images which may be copyrighted. All PLOS content is published under the Creative Commons Attribution License (CC BY 4.0), which means that the manuscript, images, and Supporting Information files will be freely available online, and any third party is permitted to access, download, copy, distribute, and use these materials in any way, even commercially, with proper attribution. For these reasons, we cannot publish previously copyrighted maps or satellite images created using proprietary data, such as Google software (Google Maps, Street View, and Earth). For more information, see our copyright guidelines: http://journals.plos.org/plosone/s/licenses-and-copyright.

4.1.    You may seek permission from the original copyright holder of Figure S1 to publish the content specifically under the CC BY 4.0 license. 

4.2.    If you are unable to obtain permission from the original copyright holder to publish these figures under the CC BY 4.0 license or if the copyright holder’s requirements are incompatible with the CC BY 4.0 license, please either i) remove the figure or ii) supply a replacement figure that complies with the CC BY 4.0 license. Please check copyright information on all replacement figures and update the figure caption with source information. If applicable, please specify in the figure caption text when a figure is similar but not identical to the original image and is therefore for illustrative purposes only.

6. We noticed you have some minor occurrence of overlapping text with the following previous publication(s), which needs to be addressed:

https://bmchealthservres.biomedcentral.com/articles/10.1186/s12913-020-05463-4

https://bmjopen.bmj.com/content/bmjopen/10/1/e034067.full.pdf

https://bmjopen.bmj.com/content/10/1/e034067.full

In your revision ensure you cite all your sources (including your own works), and quote or rephrase any duplicated text outside the methods section. Further consideration is dependent on these concerns being addressed.

Reviewers' comments:

Reviewer's Responses to Questions

**Comments to the Author**

1. Is the manuscript technically sound, and do the data support the conclusions?

Reviewer #1: Partly

2. Has the statistical analysis been performed appropriately and rigorously? 

Reviewer #1: Yes

3. Have the authors made all data underlying the findings in their manuscript fully available?

Reviewer #1: No

4. Is the manuscript presented in an intelligible fashion and written in standard English?

Reviewer #1: Yes

5. Review Comments to the Author

Reviewer #1: Thank you for the opportunity to review this manuscript. Ways to evaluate the impact of changes in health services have not been fully addressed in the literature, so this article could make important contribution to the field. However, before it gets published, I would like to suggest authors address following comments, especially the way they organized the introduction and presented the results, to increase the likelihood of having meaningful contribution to the field.

Overall suggestions:

1. Please define the terms used in the manuscript clearly. For example, how authors define “access to service”? Similar issues with “delivery pathway”, “tiered service”, and “third sector partners”. For readers who are not familiar with services in UK. It would be helpful to know the definitions of these terms.

2. Please check acronyms (e.g., CAHMS, SPA, CCG) used in the manuscript, including figures and tables, to make sure they are spelled out and explained what they mean the first time they show up, and then use acronyms thoroughly afterwards. For example, CAMHS was first spelled out in p.4 instead of p.3. Authors sometime used the full name again after the acronyms has been introduced (e.g., line 325, p.15).

3. Please use terms consistently throughout the manuscript. For example, children and young people (CYP) or adolescent? Main analysis or primary analysis?

INTRODUCTION

Overall, the links between each paragraph in the introduction were not clear, and the justification of conducting this study was not clearly elaborated. Authors described the aim of this study in the end of the third paragraph. However, the overall knowledge gap was not clearly described and summarized, so the justification of conducting this study was not very compelling in the introduction. I suggest authors reorganize the introduction to summarize the knowledge gap in a more comprehensive way, and make the link between the knowledge gap and the aim of this study more explicit. Questions regarding each paragraph were listed below:

1) In the 1st paragraph, authors described that major changes have been taking place in the organizations of mental health services, the cost and prevalence related to mental health in UK, but I was left wondering what the current unmet needs were or what the service gap looked like in UK.

2) The 2nd paragraph included concerns across high-income nations regarding adequacy of healthcare services and listed recommendations regarding how services should meet the needs of the most vulnerable. However, it was unclear to me how these concerns and recommendation were linked to the mental health services specifically. If authors tried to use this paragraph to highlight the importance of improving access, they need to make it more explicit for readers.

3) The 3rd paragraph talked about many CAMHS transferred from traditional tiered model to integrated service informed by the THRIVE model. However, it was unclear to me what existing challenges with the traditional tiered model were, how changing to integrated services could address current unmet needs, and why authors wanted to highlight THRIVE model specifically? Was it the most commonly used model across services? Please clarify.

4) The 4th paragraph talked about the evaluation of impacts and implementation of changes remained limited, the government had recommendations to ‘transform’ services, and there was significant investment in local Transformation Plans. It was unclear to me what the main idea of this paragraph was. What unmet needs caused the government to argue there’s a need to transform service? What exactly did the local transformation plans look like, and how long have these plans been implemented? What problems would exist if we didn’t evaluate impacts of changes?

5) The 5th paragraph talked about several strategies to improve services. However, it was unclear what authors’ interpretations regarding previous literature were and how these previous findings were linked to this study.

6) The 6th paragraph talked about a WHO review highlighted the importance of improving mental services and having consistent way to collect and report data. However, I don’t see how the main idea of this paragraph was connected to other paragraphs in the introduction. I would suggest authors to move it to the discussion (e.g., line 335-337, p.15).

METHODS

1. Study setting: Please provide more background information about how service transformations were taking place in Bucks and Oxon. For example, when the transformation started in each area? What were the goals of transformation in each area? Possible reasons why these two areas had service transformations while other areas did not?

2. Overall, there was not enough detail regarding qualitative data and analysis. Authors only listed types of qualitative data they collected, including observations and interviews with stakeholders. It would be helpful for readers to know the focus of observations and guiding questions they had for interviews. In addition, there was only one sentence about data analysis, which did not provide enough information for readers to evaluate the rigor of the data analysis process.

3. Line 155-159, p.7: It would be helpful for readers to see the strengths of having a mixed methods design if authors could provide several examples to elaborate how they integrated QN and QL data to achieve a more complete overall interpretation.

RESULTS

1. As authors described in the method section, there were 5 research questions. It would be easier for readers to follow if authors could consider revising the sub-headings of the result section to make the link between results and each research question more explicit. Across different parts of the result section, there were two main issues needed to be further addressed, as described below:

1) First, sometimes it was unclear for readers to see how the findings were well supported by what kinds of data. For example, in p.7, authors summarize that Bucks and Oxon had similar core components of transformation, but as a reader, it was unclear to me whether these findings were supported by any of the QN or QL data authors collected. Similarly, in p.8-9, authors described that the introduction of the SPA was perceived by staff as the core change of the CAMHS transformation, but it was unclear to me whether this finding was supported by interviews with staff OR it was supported by observations OR both. Authors also described the numbers of people accessing SPA climbed quickly without providing related data to support this finding.

2) Second, the service transformations were taking place in both Bucks and Oxon. However, sometimes authors only described findings of one area and did not provide further explanations regarding another. I understand that these two areas started service transformations at different time, so it makes sense that some findings can only be illustrated by data of Bucks. However, I still think authors should at least have several sentences to address this issue as a whole, maybe in the method section or in the beginning of the result section to act as an orientation for readers. In addition, in the facilitators and barriers section, it was also unclear to me whether the findings were support by data of two areas or only by data of one area.

2. Other suggestions:

1) Line 241-242, p.11: It seems that the “tension” is an important finding, since authors decided to have it in the abstract and the discussion. I suggest authors provide more data to support this finding.

2) P. 11-12: The findings regarding facilitators did not seem robust because the statements were not clearly supported by data.

3) P.14, table 1: Regarding the waiting time, it seemed that it increased in year 1, and then decreased in year 3. Was there any other data could provide further explanations?

DISCUSSION

Similar to the suggestion above, I suggest authors reconsider the structure of the discussion section to make it more aligned with the research questions they proposed. In addition, the discussion lacked of citing relevant literature to demonstrate how the findings of this study fit with existing knowledge and what new insights they contribute.

6. PLOS authors have the option to publish the peer review history of their article (what does this mean?). If published, this will include your full peer review and any attached files.

Reviewer #1: No

---

## [Author Response · Author response to Decision Letter 0]

1 Mar 2021

These are included, in detail, in the cover letter

---

## [Decision Letter · Decision Letter 1]

8 Apr 2021

PONE-D-20-38869R1

How does reorganisation in child and adolescent mental health services affect access to services? An observational study of two services in England

PLOS ONE

Dear Dr. Fazel,

Thank you for submitting your manuscript to PLOS ONE. After careful consideration, we feel that it has merit but does not fully meet PLOS ONE’s publication criteria as it currently stands. Therefore, we invite you to submit a revised version of the manuscript that addresses the points raised during the review process.

The original reviewer points out something that needs to be further addressed. Please take careful consideration on these comments.

We look forward to receiving your revised manuscript.

Kind regards,

Chung-Ying Lin

Academic Editor

PLOS ONE

Journal Requirements:

Reviewers' comments:

Reviewer's Responses to Questions

**Comments to the Author**

1. If the authors have adequately addressed your comments raised in a previous round of review and you feel that this manuscript is now acceptable for publication, you may indicate that here to bypass the “Comments to the Author” section, enter your conflict of interest statement in the “Confidential to Editor” section, and submit your "Accept" recommendation.

Reviewer #1: (No Response)

2. Is the manuscript technically sound, and do the data support the conclusions?

Reviewer #1: Partly

3. Has the statistical analysis been performed appropriately and rigorously? 

Reviewer #1: Yes

4. Have the authors made all data underlying the findings in their manuscript fully available?

Reviewer #1: Yes

5. Is the manuscript presented in an intelligible fashion and written in standard English?

Reviewer #1: Yes

6. Review Comments to the Author

Reviewer #1: Overall, although there are still several concerns needed to be addressed, the authors have made significant improvement regarding the clarity of the manuscript. The introduction and results are now more cohesive and focused compared to the previous version. They also added enough details to the method section.

Introduction:

p. 3 (Line 54-56): The first sentence of the introduction had some grammatical errors and included multiple ideas. I suggest the authors use more than one sentence to clearly articulate the ideas they want to convey, which would also be more reader-friendly.

p.3 (line 71-72): grammatical errors

Results:

p.13 (line 287-289): I have pointed out last time that the findings regarding facilitators did not seem robust because the statements were not clearly supported by data. However, the authors did not address this comment, and the statement “Staff showed determination…for transformation” is still not supported by any data, either from observations or interviews. Please revise.

Discussion:

I have pointed out last time that the discussion lacked of citing relevant literature to demonstrate how the findings of this study were similar or different from other work in the field of research. In the first paragraph, the authors summarized the overall findings of this study and only compared the findings with two other large-scale transformations in the NHS. In the second paragraph, the authors summarized the barriers to transformation, but did not compare the findings with any other studies. Overall, I still think the discussion section needs more work. How their findings are similar or different from other studies focusing on health service transformation? How the transformations they observed in UK are similar or different from other countries? Please elaborate and provide more details.

7. PLOS authors have the option to publish the peer review history of their article (what does this mean?). If published, this will include your full peer review and any attached files.

Reviewer #1: No

---

## [Author Response · Author response to Decision Letter 1]

9 Apr 2021

This text is also included in the Cover Letter attached. 

PONE-D-20-38869R1

How does reorganisation in child and adolescent mental health services affect access to services? An observational study of two services in England

Dear Dr Lin, 

Thank you very much for asking us to revise this manuscript and are grateful for the additional time Reviewer 1 has put into this work. We have tried to now address the final points raised and in summary are grateful for the opportunity to expand the discussion and contextualise the findings. We agree this is important, it has been partially address in our original submission (many, many moons ago) and at that point the reviewers seem to suggest these were not pertinent to our arguments and so we are delighted to have been able to update and reintroduce the broader literature for the reader. We have also addressed the request for additional evidence from qualitative interviews in the results section.

We would of course be happy to consider any further changes, and look forward to hearing from you.

Yours sincerely, 

Mina Fazel (on behalf of the authors)

Detailed Response to Reviewers comments:

Reviewer #1: Overall, although there are still several concerns needed to be addressed, the authors have made significant improvement regarding the clarity of the manuscript. The introduction and results are now more cohesive and focused compared to the previous version. They also added enough details to the method section.

Thank you for these reassuring comments. 

Introduction:

p. 3 (Line 54-56): The first sentence of the introduction had some grammatical errors and included multiple ideas. I suggest the authors use more than one sentence to clearly articulate the ideas they want to convey, which would also be more reader-friendly.

Done. This has now been changed and reads as follows:

There is heightened interest across many countries as to how mental health services can best meet the needs of their child and adolescent populations[1, 2]. The rising number of young people presenting with mental health needs is propelling services, across a number of nations, to introduce broad systemic changes.

p.3 (line 71-72): grammatical errors

Done. We hope the following has now corrected these errors:

The new model tries to enable all the agencies that are involved in a young person’s life to work together in a coherent and integrated manner

Results:

p.13 (line 287-289): I have pointed out last time that the findings regarding facilitators did not seem robust because the statements were not clearly supported by data. However, the authors did not address this comment, and the statement “Staff showed determination…for transformation” is still not supported by any data, either from observations or interviews. Please revise.

Done. Many apologies for not addressing this before. We have now added the following to this section:

Staff showed determination to deliver services aligned with the new vision for transformation, and although there were many changes that they had to manage, many described support for the overall programme of work:

“My instinct was, it was the right model with the right language with good principles and good thinking behind it and some evidence” (Staff Interview).

They agreed with the overall vision:

“I think it's creative. I think it's much more kind of proactive…there is a lot more emphasis on trying to work in the community and the proactive kind of looking at the early signs to really try and support our colleagues…in more preventative type kind of measures.” (Staff Interview)

Furthermore, staff demonstrated their commitment to the SPA with statements such as:

“it's just much more of a friendly front door to the service. (Staff Interview)

“…you can ring in at any time and you can just be reopened. You don't need a letter. You don't need a form. You don't need anybody else to do it for you. I think that's a big plus.” (Staff Interview) 

“We are meant to do two days a week in SPA, but because we've had so many calls…If somebody else is on it and I know they're inundated, I'll say 'I'll do some of them'…And definitely probably work an extra hour at least each day, and don't take a lunch.” (Staff interview)

Discussion:

I have pointed out last time that the discussion lacked of citing relevant literature to demonstrate how the findings of this study were similar or different from other work in the field of research. In the first paragraph, the authors summarized the overall findings of this study and only compared the findings with two other large-scale transformations in the NHS. In the second paragraph, the authors summarized the barriers to transformation, but did not compare the findings with any other studies. Overall, I still think the discussion section needs more work. How their findings are similar or different from other studies focusing on health service transformation? How the transformations they observed in UK are similar or different from other countries? Please elaborate and provide more details.

Done. We have tried to provide greater context to the landscape of service changes for children’s mental health and also report on studies from the current CAMHS transformation. We hope this addresses the identified need in the discussion, with which we agree. We have also had to add 28 new references. The following three paragraphs are now entirely new:

There are only a limited number of previously published studies examining broad changes to community-based child mental health services. Although implementation research has identified promising strategies to improve services including: enhanced engagement to retain families in services [26]; improved training and support for evidence-based practices [27]; and expanded measurement and feedback systems to monitor services in real time, actual evaluation data remains limited [28]. The only studies identified in the last decade that have examined broad system change to improve demand and capacity in child and adolescent mental health services have included studies on the positive impact of introducing a ‘Choice and Partnership’ Approach. These services conduct an initial appointment to reach a shared understanding of patient and family needs with a range of options offered following the meeting [29-31]. Another study evaluated ‘Shared care mental health care’ (with primary care) [32] demonstrating how this model increases access to care as well as decreasing demand on services and a final study of integrated behavioural health care into primary health care systems showed some positive improvements in symptoms [33]. There are therefore some broad similarities with the current ‘transformation’ of CAMHS in England with a focus on integration and better shared decision-making.

There have been a number of evaluations of youth mental health service change for youth aged up to 25 years, in particular of the Australian headspace initiative [34-37]. Although relevant, the target age range is different. Numerous studies describe specific interventions introduced into child mental health systems, including: family check-up [38]; wraparound services for serious emotional disturbance [39]; parenting interventions [40]; development of interim services whilst awaiting mental health services [41]; free counselling support [42]; telephone-based treatments [43]; assertive outreach teams [44]; and early intervention in psychosis services [45, 46]. 

A few articles have been published on the recent, broader system changes in the UK, primarily on the introduction of the Children and Young People’s Improving Access to Psychological Therapies (CYP-IAPT) changes [47]. These are changes that have preceded the current CAMHS transformation, and have focused on ensuring that there are both more CAMHS practitioners and that these practitioners are trained to deliver evidence-based psychological therapies to child and adolescent populations [48]. The studies reporting on these changes do share similar findings on operational difficulties, the need for stakeholder involvement and the importance of leadership, although actual service-related measures, such as in this study, have not been reported [49, 50]. The current CAMHS transformation has been described in some publications [51] with qualitative work conducted on key stakeholders [52], although some concerns raised that insufficient young people and parents have been included [53]. Quantitative data on the impact on access to services and efficiency in newly ‘transformed’ CAMHS service delivery has not been identified in any published studies to date- reflecting the recency of these changes. This study is therefore a timely evaluation of CAMHS provision in the context of transformation.

---

## [Editor Report · Decision Letter 2]

13 Apr 2021

How does reorganisation in child and adolescent mental health services affect access to services? An observational study of two services in England

PONE-D-20-38869R2

Dear Dr. Fazel,

We’re pleased to inform you that your manuscript has been judged scientifically suitable for publication and will be formally accepted for publication once it meets all outstanding technical requirements.

Kind regards,

Chung-Ying Lin

Academic Editor

PLOS ONE

Additional Editor Comments (optional):

Thank you for all the hard efforts and I feel that you have done a good job!
---

## [Editor Report · Acceptance letter]

23 Apr 2021

PONE-D-20-38869R2 

How does reorganisation in child and adolescent mental health services affect access to services? An observational study of two services in England 

Dear Dr. Fazel:

I'm pleased to inform you that your manuscript has been deemed suitable for publication in PLOS ONE. Congratulations! Your manuscript is now with our production department. 

Kind regards, 

on behalf of

Dr. Chung-Ying Lin 

Academic Editor

PLOS ONE